# The 'Double Helix' model of quality monitoring: Risk mapping of quality management system during initial ISO 15189 Implementation in a medical laboratory

**Shuzhe Yang**[1]◉, **Yali Zhou**[2]◉, **Chunyuan Wang**[1]◉, **Mengjun Luo**[1]*

1 Department of Clinical Laboratory, Chengdu Women's and Children's Central Hospital, School of Medicine, University of Electronic Science and Technology of China, Chengdu, China, 2 China National Accreditation Service for Conformity Assessment, Beijing, China

◉ Shuzhe Yang, Yali Zhou, Chunyuan Wang share first authorship
* 462816877@qq.com

## Abstract

### Objective

Analyze and compare the characteristics of nonconformities (NCs), root causes, and corrective actions generated from internal assessments and ISO 15189 external assessments in a medical laboratory, identify high-risk points and evaluate the role and contribution of different assessments to the medical laboratory's quality performance.

### Methods

A total of 35 NCs from internal assessments and 67 NCs from external assessments were documented in a medical laboratory between 2021 and 2024. The NCs were categorized according to clause requirements, while root causes and corrective actions were classified based on their content. The recurrence rate of NCs, effectiveness of corrective actions and characteristics of recurring NCs were statistically analyzed. The flow relationships were shown by Sankey diagrams. The laboratory quality monitoring model was illustrated by a double-helix diagram.

### Results

In internal assessments, the top three NC categories were Examination processes (n = 8, 22. 86%), Personnel management (n = 7, 20. 00%), Document and record control (n = 6, 17. 14%);in external assessments the top categories were were Examination processes (n = 19, 28. 36%), Document and record control (n = 10, 14. 93%), Others(Evaluation, Complaints, Information system, Risk Management) (n = 8, 11. 94%), Personnel management (n = 6, 8. 96%). Regarding root causes,

**Data availability statement:** All relevant data are within the manuscript and its Supporting Information files.

**Funding:** The author(s) received no specific funding for this work.

**Competing interests:** The authors have declared that no competing interests exist.

the most frequent in internal assessments were Personnel Negligence (n = 14, 40. 00%), Training deficiencies (n = 11, 31. 43%);in external assessments, the tops were Training deficiencies (n = 27, 40. 30%), Document and Record Deficiencies(n = 15, 22. 39%). The The most frequently implemented corrective action across both assessment types was Personnel training(internal [n = 35, 47. 29%] and external [n = 67, 41. 36%]). The numbers of Managment NCs, Technical NCs, total NCs, and Corrective Actions in external assessments were significantly higher than in internal assessments.

## Conclusion

The high-risk points primarily lie in Examination processes, Document/record and Personnel management during the initial implementation of the ISO 15189. External assessments help identify deviations, contributing to quality performance improvement. Internal assessments enable continuous monitoring of quality issue corrections, supporting the ongoing enhancement of the QMS. The 'Double Helix' Model of Quality Monitoring ensures the stability and accelerated advancement of the quality management system in the medical laboratory.

## Introduction

The data generated from the laboratories are becoming decision engines [1]. So medical laboratories play a critical role in patient diagnosis and treatment. The defining feature of the Industry 4. 0 era is the immediate flow of information and the reactions to the changes without delay [2]. The importance of timely accurate test results from medical laboratories will continue to grow in future healthcare. This new paradigm requires medical laboratories meet higher quality standards to keep pace with the times. The QMS in a medical laboratory is complex and extensive. It encompasses personnel, instruments, reagents, methodologies, and environmental conditions horizontally. It also spans the entire testing process (pre-analytical, analytical, and post-analytical phases) vertically [3–5]. ISO 15189 is a QMS specifically designed for medical laboratories, regularly updated according to reflect advances in medicine and management science [6–9]. ISO 15189 QMS promotes the guidelines for periodical, systematic monitoring of processes and identifying areas which might need improvement [10]. The ISO 15189 accreditation project has been widely adopted in European countries and sub-Saharan Africa [11,12]. The latest version is ISO 15189:2022 emphasizing risk assessment implementation [3]. In the era of evidence-based medical practice, risk management is an essential quality improvement activity that must be integrated in all testing processes [13].

The initial phase of implementing a QMS is often unstable, characterized by vulnerabilities and challenges. This period is also critical in determining whether the system will be successfully established. As Laila O. AbdelWareth et al. 's research indicated team dynamics and work culture are still evolving during this phase due

to the new environment [14]. Accurately identifying high-risk points and implementing effective preventive measures can help laboratories navigate the transition smoothly. Internal and external assessments are two distinct assessment methods. Both are conducted based on ISO 15189 standards and national regulations, but they differ in personnel, cost, and frequency. Internal assessments are performed by laboratory staff [4], with fixed labor costs and are conducted annually. External assessments, however, are carried out by assessors trained by CNAS with higher costs due to assessment fees [15]. In the first three years, external assessments are conducted annually, transitioning to biennial assessments after the third year in China.

To understand the impact of different assessment methods on medical laboratories during the initial stages of QMS implementation in China, this study compared the characteristics of NCs, recurrence rates, and corrective actions between internal assessments and external assessments. By identifying high-risk points and weaknesses, the study provides insights to help medical laboratories achieve a smoother transition.

## Methods

### Laboratory settings

The study was conducted in the Clinical Laboratory at specialized women and children's hospital in Chengdu, China. This is a 1840-bed hospital. The Clinical Laboratory offers routine and STAT tests. It has several departments including clinical chemistry, immunology, microbiology, molecular diagnostics. body fluids, hematology. The laboratory conducts a comprehensive internal audit annually in accordance with the requirements of ISO 15189 from 2021. The laboratory received ISO 15189 accreditation in 2022. The accreditation was subsequently maintained following a surveillance review in 2023 and a successful first reassessment in 2024.

### Data collection

After gaining approval from the laboratory administrators, NCs were collected from 4 internal assessments and 3 external assessments retrospectively from January 2021 to December 2024. All nonconformities (NCs) identified during both assessments were included in the study. All internal assessors possessed valid certification. External assessments were organized by the China National Accreditation Service for Conformity Assessment (CNAS).

All identified NCs were primarily classified according to their corresponding clauses within the ISO 15189:2022 standard. Each root cause of NC was subsequently assigned to one of five predefined categories:Document and Record Deficiencies(Absence, error, obsolescence, or inaccessibility of quality management system documents; or non-standard, incomplete, or non-traceable records), Training Deficiencies(Failure of the laboratory to conduct required training, failure of personnel to receive necessary training, or failure of training to achieve its intended outcomes), Process Deficiencies(Poorly designed documented operational procedures that are difficult to execute, misaligned with the actual work environment, overly complex, or prone to operational error), Personnel Negligence(Momentary lapses or habitual actions leading to accidental deviations from established correct procedures, or personnel "inattention," "omission," "lack of awareness," or "inability to perform."). Resource Constraints(Insufficiency of human resources, equipment, reagents and consumables, facilities, or information systems).

Corrective actions were categorized based on the primary activity undertaken. Each Corrective Action of NC was subsequently assigned to one of seven categories:Personnel Training, Document Modification, Supervision and Inspection, Assessing the Impact of NCs, LIS Modification, Supplementary Records, Others (encompassing all measures not covered by the preceding six categories).

A dual-review process was used to ensure objectivity and consistency. Two assessors were trained and experience in ISO15189:2022 QMS. They independently evaluated and categorized each NC according to the predefined classification framework. During the evaluation, both assessors were blinded to each other's judgments. All initial classifications were

then compared. For any result where the two assessors' categorizations disagreed, a structured consensus discussion was held to review the evidence and rationale. If consensus could not be reached through discussion, a third senior quality manager (the laboratory's quality director who was not involved in the initial review) was consulted to make a final, binding adjudication.

## Data analysis

To enable the quantitative assessment of quality improvement activities, two core process quality indicators were employed in this study: NC Recurrence Rate and Annual Cumulative Corrective Action Effectiveness Rate. These indicators are designed to monitor the effectiveness of implemented corrective actions [16]. Their definitions and computational methods are detailed below. 1. Annual NC Recurrence Rate =(No. of recurrent NCs in the current year/ Total No. of NCs in the same assessment type in the current year)× 100%. Recurrent NC:the NC related to the same specific clause of the ISO 15189 standard as a previously documented NC. The previously documented NC had been fully corrected through corrective actions and formally closed within the quality management system. 2. Annual Cumulative Corrective Action Effectiveness Rate=(1−No. of recurrent NCs in the current year/Cumulative total of NCs in the same assessment type up to the end of the previous year)× 100%. Cumulative total of NCs in the same assessment type up to the end of the previous year: aggregate number of all NCs documented within the same assessment type from the start of the recording period to the end of the year preceding the calculation year. 3. Corrective Action Effectiveness Rate=(Number of corrective actions corresponding to NCs that were not Recurrent/ Total implemented corrective actions) × 100%.

The flows among NCs, root causes and corrective actions were illustrated by Sankey diagrams in internal and external assessments. Sankey diagrams were generated using OriginPro 2024. The process involved the following steps: defining the source and target nodes, with source nodes in this study being the categories of NCs and root causes, and target nodes being the root causes and corrective actions, respectively; quantifying the frequency of occurrences from each source node to the respective target nodes; and establishing links between the sources and target nodes. The Sankey diagrams visually illustrate the quantitative relationships between the source and target nodes.

The double-helix model was designed in PowerPoint software(12. 1. 0. 24034). Statistical analysis was performed to compare distributions and trends of NC types, root causes, and corrective actions.

## Statistical methods

Statistical analysis was analysed with SPSS 25. 0 software. The Mann-Whitney U test was used to compare the distributions of count data. This nonparametric test was chosen because the data were derived from independent samples with a limited sample size and were found to deviate from a normal distribution. The rank-biserial correlation coefficient was reported as the effect size. Due to the presence of independent samples with small expected cell counts (<5), Fisher's exact test was used to compare proportions between categorical variables. The **Risk Ratio** along with its 95% confidence interval was reported as the effect size. All tests were two-tailed, $P < 0.05$ considered statistically significant.

## Results

### Distribution of NCs, root causes, major corrective actions in internal and external assessments

S1Table showed the distribution of NCs according to ISO 15189 requirements in internal and external assessments. The top three NC categories were Examination processes(n=8, 22. 86%), Personnel management (n=7, 20. 00%), Document and record control (n=6, 17. 14%) in internal assessments, and Examination processes (n=19, 28. 36%), Document and record control (n=10, 14. 93%), and Others (n=8, 11. 94%), Personnel management (n=6, 8. 96%)in external.
S2 Table showed the top root causes were Training Deficiencies (n=14, 40. 00%), Training deficiencies (n=11, 31. 43%) in internal assessments, and Training Deficiencies(n=27, 40. 30%), Document and Record Deficiencies(n=15, 22. 39%),

Personnel Negligence(n = 14, 20. 90%) in external assessments. S3 Table showed the top implemented corrective actions were Personnel training(internal [n = 35, 47. 29%]and external [n = 67, 41. 36%]), the second in internal assessments were Supplementary records(n = 28, 37. 84%) and Document modification(n = 23, 14. 20%) in external assessment Fig 1.

### Analysis of the NCs characteristics in internal and external assessments

Table 1 showed the comparison and statistical analysis of the NCs in internal and external assessments. The NO. of Managment NCs, Technical NCs, total NCs and Corrective Actions in external assessments were significantly higher than internal assessment($p < 0. 05$). No significant differences were observed in New occurrences NCs, Repeats NCs, Recurrence rate per year and Corrective action effectiveness Rate per year in internal and external assessments.

### Analysis of major corrective actions for NCs in internal and external assessments

Table 2 showed the distribution and the statistical analysis of corrective actions in internal and external assessments. The effectiveness Rate in personnel training was higher in external assessments than internal assessments, while the effectiveness Rates in Document modification and Supervision/ inspection were lower in external assessments.

### The flow relationships between NCs and root causes, as well as root causes and corrective actions in Sankey diagrams

In Fig 2 From left to right in the Sankey diagram, the columns represent: NCs contents, categories of root causes, and frequency of occurrence. In the internal assessment, the flow from "Personnel" NCs to "Personnel Negligence" appeared most frequently (5 times). Over half of the flows between NC types and different root causes occurred only once. In the

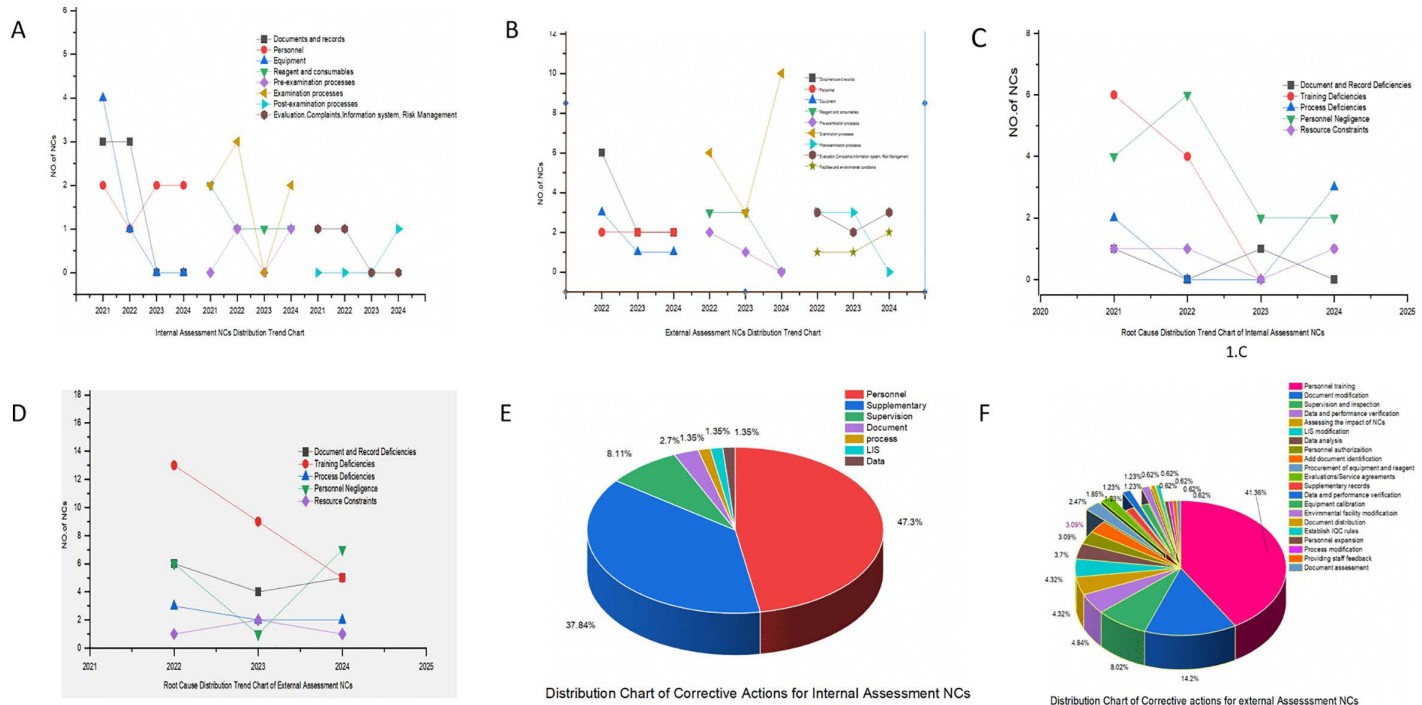

**Fig 1. NCs Distribution Trend Chart in Internal and External Assessments (A, B); Root Causes for NCs Distribution Trend Chart in Internal and External Assessments (C, D); Corrective Actions for NCs Distribution Chart in Internal and External Assessments (E, F).**

**Table 1. Comparison and Statistical analysis of the NCs characteristics in internal and external assessments.**

| variables | Internal assessment | | | | External assessment | | | P | Effect size |
|---|---|---|---|---|---|---|---|---|---|
| | 2021 | 2022 | 2023 | 2024 | 2022 | 2023 | 2024 | | |
| Total NCs(N) | 14 | 11 | 3 | 7 | *29* | *18* | *20* | 0. 034 | −0. 80 |
| Management NCs(N) | 2 | 1 | 1 | 0 | *8* | *2* | *4* | 0. 048 | −0. 75 |
| Technical NCs(N) | 12 | 10 | 2 | 7 | *21* | *16* | *16* | 0. 032 | −0. 80 |
| New occurrences(N) | 14 | 10 | 2 | 5 | *29* | *13* | *13* | 0. 154 | −0. 54 |
| No. of recurrences (N) | / | 1 | 1 | 2 | */* | *5* | *7* | 0. 076 | −0. 79 |
| Annual NC Recurrence Rate*(%) | / | 9. 09 | 33. 33 | 28. 57 | */* | *27. 78* | *35. 00* | 0. 564 | −0. 26 |
| ACCAER**(%) | / | 92. 86 | 96. 00 | 92. 86 | */* | *82. 76* | *85. 11* | 0. 076 | −0. 80 |
| Corrective Actions(N) | 31 | 22 | 7 | 14 | *67* | *45* | *50* | 0. 034 | −0. 80 |

*Annual NC Recurrence Rate =(No. of recurrent NCs in the current year/ Total No. of NCs in the same assessment type in the current year)× 100%

**Annual Cumulative Corrective Action Effectiveness Rate=(1−No. of recurrent NCs in the current year/Cumulative total of NCs in the same assessment type up to the end of the previous year)× 100%

external assessment, the flow from "Examination Processes" NCs to "Training Deficiencies" was the most frequent (9

**Table 2. Comparison and statistical analysis of major corrective actions for NCs in internal and external assessments.**

| Sub-dimensions | NO. of Corrective Actions (N) | | Corrective Action Effectiveness Rate* (%) | | p | Effect size |
|---|---|---|---|---|---|---|
| | Internal assessment | *External assessment* | Internal assessment | *External assessment* | | |
| Personnel training | 35 | *67* | 88. 57 | *94. 73* | 1. 000 | 1. 023[0. 879-1. 191] |
| Document modification | 2 | *23* | 100 | *91. 30* | 1. 000 | 1. 095[0. 965-1. 242] |
| Supervision and inspection | 6 | *13* | 100 | *76. 92* | 0. 517 | 1. 375[0. 965-1. 751] |
| Assessing the impact of NCs | 0 | *7* | / | *71. 42* | | |
| LIS modification | 1 | *7* | 100 | *71. 42* | 1. 000 | 1. 400[0. 876-2. 237] |
| Personnel authorizaition | 0 | *5* | / | *80. 00* | | |

*Corrective Action Effectiveness Rate=(Number of corrective actions corresponding to NCs that were not Recurrent/ Total implemented corrective actions) × 100%.

*:statistical analysis of the Corrective Action Effectiveness Rate between internal and external assessments

times). Similarly, over half of the flows occurred only once. Compared with the diagram from the internal assessments, more types of NCs are involved and more diverse and complex flows to root causes are exhibited in the external assessments. Fig 3 From left to right, the columns represent categories of root causes, types of corrective actions, and frequency of occurrence. In the internal assessments, the flow from Personnel Negligence to the Personnel Training was the most frequent (14 times). In the external assessments, the flow from "Training Deficiencies" to "Personnel Training" occurred most frequently (27 times). Compared with the internal assessments diagram, the external assessments involved a greater variety of corrective actions, with identical root causes flowing to a wider range of different corrective actions.

## Analysis and the characteristics of recurrent NCs

Table 3 showed No. of Corrective Actions in the recurrent NCs from external assessments was significantly higher than that from the internal assessments. No significant differences were observed in Recurrence Rate and the Recurrence interval time.

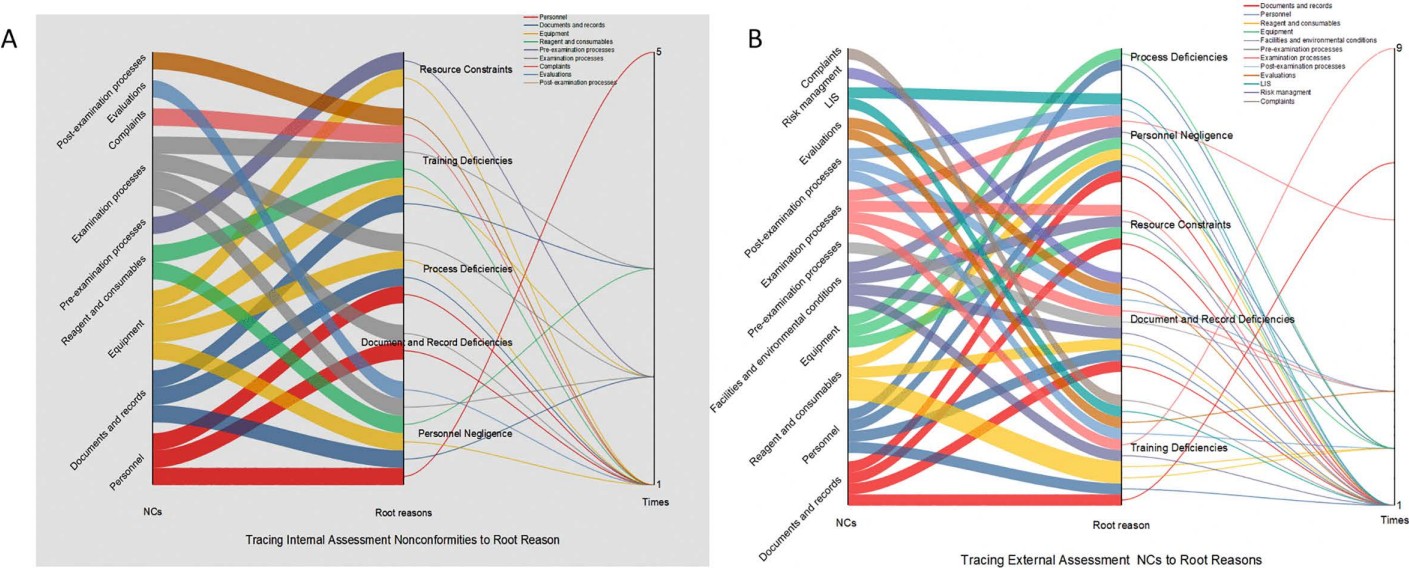

**Fig 2.** The Sankey diagram tracing from NCs Contents to Root Causes in Internal assessments (A); The Sankey diagram tracing from NCs to Root Causes in External assessments (B).

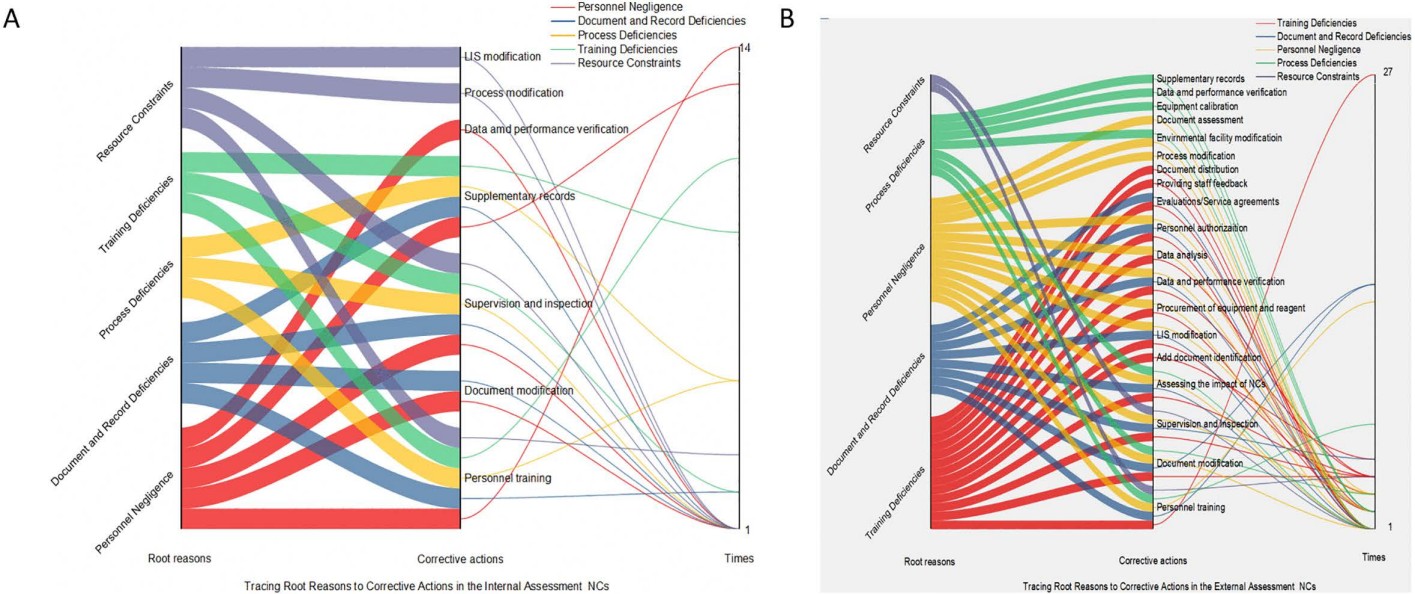

**Fig 3.** The Sankey diagram tracing from Root Causes to Corrective Actions in Internal assessments (A); The Sankey diagram tracing from Root Causes to Corrective Actions in External assessments (B).

**Table 3. Statistical analysis and the characteristics of recurrent NCs.**

| | Reqirments(type/(N)) | Root causes(type/(N)) | No.of Corrective Actions(Mean±SD) | Recurrence Rate(%) | Recurrence interval(year) |
|---|---|---|---|---|---|
| Internal assessment | Reagent and consumables(3) Verification of examination methods (1) | Personnel Negligence(3) Training Deficiencies(1) | 2.00+0.00 | 14.28 | 1.25+0.50 |
| *External assessment* | *Examination processes(5) Post-examination processes(2) Reagent and consumables(2) Personnel(1) Equipment (1) Documents and records(1)* | *Training Deficiencies (4) Personnel Negligence(3) Document and Record Deficiencies(2) Resource Constraints(2) Process Deficiencies(1)* | *2.50+0.52* | *25.53* | *1.50+0.52* |
| P | | | 0.083 | 0.249 | 0.398 |
| Effect size | | | -0.43 | 0.557 [0.208, 1.492] | -0.21 |

## The 'Double Helix' model of quality monitoring

The results revealed that in the initial phase of QMS establishment, external assessments detected a higher number and a broader distribution of NCs compared to internal assessments, exerting a wider chain and stronger supervisory influence on the QMS. However, after three years of operation, internal assessments began to identify NC types that were not captured by external assessments. The "chain" of internal assessment widened, and its supervisory influence strengthened. Together, internal and external assessments jointly ensure that the quality architecture climbing in a continuously dual-helix pattern. Both assessments involve root causes and corrective actions, which determine whether NC will recur. The two chains are linked by root causes and corrective actions, then a stable double-helix could be presented (Fig 4).

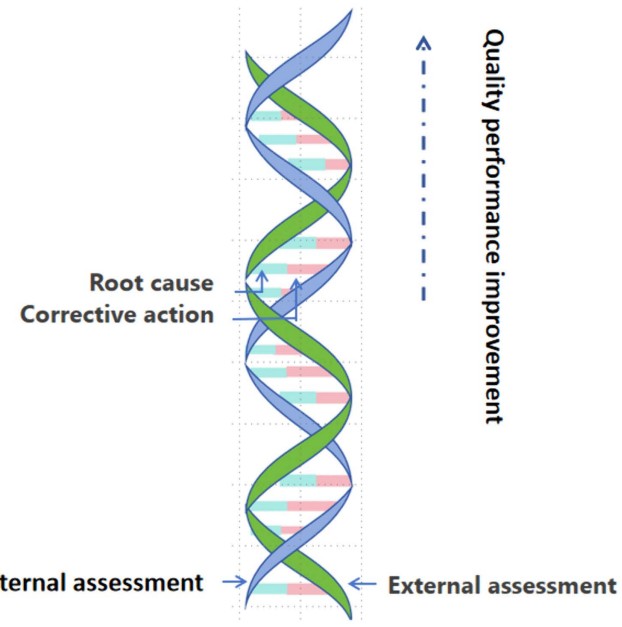

**Fig 4. The 'Double Helix' model of quality monitoring in Internal and External assessments.**

## Discussion

Each medical laboratory should strive to establish and maintain a QMS because laboratory results play a vital role in disease diagnosis and treatment protocols [17]. As a key intervention to strengthen laboratory services, QMS implementation requires proper planning to avoid various difficulties during the process [18]. Therefore, the timely and precise identification of systemic high-risk points is essential to mitigate implementation challenges.

The results showed that during the initial establishment of the QSM, the high-risk points were in the examination processes, document/record control and personnel management. In the research on ISO 15189-accredited laboratories in Hong Kong, the results were largely aligned with the only discrepancy observed in personnel management. This discrepancy may be related to factors such as the duration of quality system operation and the number of laboratory staff. Among the requirements for examination processes, the area with the highest number of nonconformities (NCs) was internal quality control (IQC). The aim of IQC is to monitor the examination process to prevent the generation of erroneous information [19]. Therefore, assessors place particular emphasis on IQC performance. Current challenges in IQC implementation in China include the following [20]:(I)IQC procedures based on different quality control materials; (II)Selecting appropriate quality control materials;(III)Determining method performance (scientifically selecting TEa, CV, and Bias); (IV) Establishing statistical quality control strategies based on method performance and patient risk; (V)Strategies for addressing out-of-control situations; (VI)IQC for qualitative testing items. The NCs identified in our laboratory correspond to all six of these challenge areas, so addressing these risk points is key to improving our IQC quality. For a laboratory in the initial phase of establishing its QMS, there are significant inherent challenges in developing a document and record control system from scratch. Furthermore, our testing processes must not only comply with ISO 15189 requirements but also align with Chinese national standards, regulations, and the practical operational needs of clinical work. Personnel management procedures contains training, competency assessment, and authorization. Personnel management procedures **contain** training, competency assessment, and authorization. It is a systematic and flexible evaluation system that does not rely solely on training. The supervisor should focus not only on satisfactory acquisition of the knowledge, skills, but also evaluating the capability, professionalism and potential of the personel [21]. Internal assessments generated fewer NCs in the areas of Evaluation, Complaints, Information system and Risk Management than external assessments. This may be due to the flexible nature of evaluations and complaints, which requires assessors to adeptly integrate quality requirements with actual laboratory operations. While risk management has been given greater emphasis in ISO 15189:2022, internal assessors are mostly engaged in technical management work and may not have fully understood the updated requirements in the short term. Notably, NCs related to examination processes showed a rebound, further indicating instability during the initial QMS implementation phase. To ensure the stable quality level, it would be beneficial to implement quarterly or semi-annual monitoring and review of the corrective action outcomes. The number of NCs identified in external assessments showed high variability without a sustained downward trend, whereas internal assessments exhibited a marked initial decline followed by a slight rebound. In external assessments, no increasing trend was observed in the second year of operation (with five categories even showing a decrease). However, in the third year, two categories displayed an increase, indicating instability in the QSM. A post-accreditation "risk period" appears to exist, during which vigilance in quality activities and internal oversight may diminish. Consequently, laboratories that have implemented ISO 15189 for 1–2 years require enhanced monitoring following consecutive successful assessments, particularly within examination processes. The detection efficiency of internal assessments was initially lower, as many NCs identified externally were missed internally. Nevertheless, in the third year, some NCs were detected internally but not externally, suggesting that internal assessments initially served a supplementary function, with their complementary value becoming more prominent after three years. Therefore, although influenced by factors such as personnel competence and objectivity, internal assessments remain critically important during the first 3–4 years of ISO 15189 implementation and may yield progressively greater positive impact as the QMS matures.

Training deficiencies were the most common root cause in both internal and external assessments. Aligning with these findings, training was the predominant corrective action. For the medical laboratory personnel, training encompasses situational awareness of management system standards [22]. The dissemination of QMS requirements, implementation of technical requirements, and formulation of corrective actions all depend on robust training. We propose a systematic training optimization framework. This framework contains four actionable pillars: content comprehensiveness, trainer competency, format effectiveness, and assessment validity. The training needs to be repeated, with increased frequency for high-risk areas. Documentation and Record deficiencies were the second top root cause in external assessments, but fifth in internal assessments. Document modification (23 times) ranked second among corrective actions and was higher in external assessments. This difference reflects the varying capabilities of the two assessment types in identifying documentation issues. Since internal assessors are the main authors and approvers of documents, these documents naturally become blind spots in internal assessment. To effectively mitigate Document and Record Deficiencies, a comprehensive operational plan comprising four core measures is proposed. First, strictly enforce an "author recusal" and "cross-review" system. Second, conduct periodic systematic document and record review campaigns. Third, regularly engage external experts to perform document and record evaluations. Finally, establish a fully traceable lifecycle management system covering document creation, approval, issuance, review, and obsolescence.

The NO. of Managment NCs, Technical NCs and total NCs identified in external assessments were significantly higher than in internal. The difference in competency between internal and external assessors may be the primary reason: external assessors have extensive experience in laboratory quality management and have undergone multiple CNAS training sessions and assessments, giving them comprehensive mastery of QSM; whereas our department's internal assessors received training from various institutions, typically lasting only 2–4 days, with only one assessment that was less rigorous than that for external assessors. Medical laboratories need to ensure that the internal assessment checklists could cover relevant compliance requirements in all relevant clauses [21]. The NO. of Corrective Actions in external assessments were significantly higher than internal. The high volume of corrective actions indicates both the complexity of the NCs and depth of the external assessments findings. It is suggested that external assessment are more effective at uncovering NCs and improve laboratory quality by revealing areas for enhancement in the QMS operation. These findings support the recommendation that laboratories in the initial stages of establishing QSM should regularly conduct external assessments to monitor their quality systems. Ian G. Wilson' reserch showed similar proportions of NCs between internal and external assessments in a public health laboratory in United Kingdom [23]. With the prolonged establishment of the quality system, the gap in influence between internal and external assessments on lab quality will be narrowed. The number of corrective actions was significantly higher in external assessments. The greater number of NCs raised from external assessments is the main reason. Additionally, laboratories tend to implement multiple corrective actions to ensure their effectiveness and maintain accreditation status. No significant difference was observed in the corrective action effectiveness rate between the two assessment types, though this result may have been influenced by the small sample size. The data indicated a higher personnel training effectiveness rate in external assessments. Training conducted by the external assessors serves as an effective approach. During external assessments, assessors clarify the requirements of relevant clauses and highlight high-risk implementation points based on the specific context of the laboratory. This process helps laboratory staff gain an in-depth understanding of these NCs. Subsequently, within a short period (approximately one week) after the external assessment, relevant personnel conduct training for laboratory staff based on the learnings from the external assessors.

The analysis identified the most significant cause of NC recurrence was insufficient scope of corrective action implementation. (I) Corrective measures were primarily applied only within the group where the NC was initially identified (e.g., Reagents and Consumables). Due to sampling limitations or variations in assessor focus, NCs were not detected in other groups. (II) Although some corrective actions included a review of similar issues in other groups, the verification methods were often superficial and review was biased. There may have been a tendency to select well-performing groups or

assays to expedite NC closure (internal quality control failures in Examination Processes). (III) Inadequate verification of training effectiveness: Recurrence rates were highest in Personnel Negligence and Training Deficiencies. The current training model: the group leader designs and delivers the training, followed by an immediate written or oral test. Trainer competence, training methodology, and the timing/format of assessment all critically influence outcomes. We recommend that training be delivered face-to-face, repeatedly. The trainer has strong system knowledge and communication skills. The assessments should be conducted separately, not on the same day as training. (IV) Cross-departmental coordination gaps: some corrective actions were intersects with other hospital departments. They were implemented without considering the workflows of these departments(e. g. , equipment management). The followings are recommended during the implementation of corrective actions. Treat the quality element linked to any NC as a high-risk item for all professional groups. A complete (non-sampling) review of compliance with the relevant clause should be conducted across all groups. The reviewers are leadership or personnel independent of any group's interests, who possess a thorough understanding of quality-system requirements. Checklist templates should be designed to correspond directly to ISO 15189 requirements, allowing for sequential verification of each clause. Ensure the laboratory procedures explicitly consider the impact of related departmental policies.

The Sankey diagram revealed that the NCs identified in external assessments involve more requirement clauses and more diverse corrective actions compared to internal assessments. This discrepancy may stem from the fact that the medical laboratory was in the initial phase, where both staff competency and overall quality capabilities were still developing. Consequently, the effectiveness of internal assessments was weaker than that with well-established quality systems.

At the initial stage, the core function of external assessments is to identify quality deviations to ensure the overall compliance of the system. It effectively guides the quality framework in a continuous upward spiral. In contrast, the function of internal assessments focuses on horizontally extending supervision to achieve comprehensive risk prevention and control. It facilitates the thorough correction of NCs. This dual-helix model provides more stable safeguarding of the quality system than any single monitoring mechanism. Root causes and corrective actions drive substantive improvements in the quality system. They determine whether the "dual-helix" model remains stable and resistant to collapse.

To align with the sustainable development of healthcare systems, future updates to ISO 15189 will likely incorporate sustainability requirements [24]. Medical laboratories must continuously improve quality standards to adapt to the sustainability and digital era. The studies in Kenya and India have found that the implementation of the ISO 15189 lead to improvements in the pre-analytical, analytical, and post-analytical phases of testing, particularly in terms of sample management and External Quality Assessment (EQA) performance [25–27]. For developing or less economically developed regions, the ISO 15189 QSM serves as an effective tool for enhancing laboratory quality. We recommend adopting a "dual-helix" model to ensure the stability and accelerated advancement of the quality system.

This study uses data from a single center, the generalizability is constrained by factors specific to our laboratory setting. Such factors include: (I) laboratory scale and resource allocation, (II) service population characteristics, (III) personnel competency, (IV) QMS maturity level, and (V) the level of automation and informatization in testing processes. Although this study employed the internationally recognized ISO 15189 standard as an evaluation framework, variations in the interpretation and implementation of this standard may affect the generalizability across different countries, regions, and institutions. Additionally, this study may not have fully captured all NCs within the quality system due to the sampling methods used or variations in auditor competency. The depth of auditors' (particularly internal auditors') understanding of ISO 15189 clauses and their tolerance thresholds for deviations in the quality system may influence the data. Certain subgroup analyses may have been underpowered due to low event counts. This could have prevented the detection of statistically significant differences. In summary, when other laboratories applying the indicator framework and analytical methods developed in this study, they should conduct a systematic evaluation and make necessary adaptations based on their specific conditions.

Future research could further explore the following directions: extending the observation period to capture long-term dynamics in quality system evolution; quantifying changes in fundamental laboratory conditions (e. g. , personnel structure, equipment investment, informatization level) and incorporating them as covariates in the analysis; and employing multivariate models to assess the impacts of these conditional changes on the quality system. Collect quality data from laboratories with different characteristics and from different geographic regions. Identify meaningful laboratory quality indicators. Leverage artificial intelligence to screen sensitive quality indicators, facilitating precise identification of deviations within the quality system. Establish an automated AI-powered medical laboratory quality assessment system which can perform impartial, objective, and comprehensive evaluation.

## Supporting information

**S1 Table. Distribution of NCs in internal and external assessments.**
(DOC)

**S2 Table. Distribution of root causes in internal and external assessments.**
(DOC)

**S3 Table. Distribution of major corrective actions in internal and external assessments.**
(DOC)

## Author contributions

**Data curation:** Chunyuan Wang.

**Writing – original draft:** Shuzhe Yang.

**Writing – review & editing:** Yali Zhou, Mengjun Luo.

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
