## [Decision Letter · Decision Letter 0]

14 Nov 2025

Dear Dr. LUO,

Thank you for submitting your manuscript to PLOS ONE. After careful consideration, we feel that it has merit but does not fully meet PLOS ONE’s publication criteria as it currently stands. Therefore, we invite you to submit a revised version of the manuscript that addresses the points raised during the review process.

We look forward to receiving your revised manuscript.

Kind regards,

Arne Johannssen

Academic Editor

PLOS ONE

Journal Requirements:

2. We note that your Data Availability Statement is currently as follows:

“All relevant data are within the manuscript and its Supporting Information files.”

4. Please ensure that you refer to Figures 1 and 2 in your text as, if accepted, production will need this reference to link the reader to the figure.

**Additional Editor Comments :**

Please carefully address the concerns raised by both reviewers.

Reviewers' comments:

Reviewer's Responses to Questions

**Comments to the Author**

1. Is the manuscript technically sound, and do the data support the conclusions?

Reviewer #1: Yes

Reviewer #2: Yes

2. Has the statistical analysis been performed appropriately and rigorously?

Reviewer #1: Yes

Reviewer #2: Yes

3. Have the authors made all data underlying the findings in their manuscript fully available?

Reviewer #1: Yes

Reviewer #2: No

4. Is the manuscript presented in an intelligible fashion and written in standard English?

Reviewer #1: Yes

Reviewer #2: No

Reviewer #1: The study provides a valuable, data-driven exploration of quality monitoring during the early stages of ISO 15189 implementation. The manuscript technically sound, and do the data support the conclusions and the statistical analysis been performed appropriately and rigorously. Moreover, the data is available without restrictions to any user. The authors successfully highlight high-risk areas, the role of assessments, and the importance of training and documentation. However, single-laboratory-scope, internal bias risks, and limited recurrence analysis suggest areas for improvements in future research.

Reviewer #2: 1. Clarity and Language

There are a lot of typographical, grammatical, and formatting mistakes in the document (such as "the datas," "the potential," and "supervision/inspectionin"). Sometimes sentences are lengthy and complicated, making them harder to read. To improve clarity and flow, please have a professional scientific editor or native English speaker perform thorough English language editing.

2. "Double Helix" Conceptual Model

Although the "Double Helix" concept is an intriguing and inventive parallel, it is not well defined. Describe the model's theoretical underpinnings and how internal and external assessments work together as "strands" of ongoing quality improvement. Provide a schematic picture of the model to illustrate its functioning within the framework of ISO 15189 implementation.

3. Methodological Details

Make clear if inclusion/exclusion criteria were used or if all nonconformities (NCs) found between 2021 and 2024 were examined. Give a more thorough explanation and rationale, preferably with references, for the formulas used to determine the Recurrence Rate and Corrective Action Effectiveness Rate. Describe the methods used to assure data reliability, such as independent reviewers and double-checking NC classification. Talk about the limits that come with researching a specific institution and how this could impact generalizability.

4. Data Presentation and Statistics

a. When applicable, assumptions, sample distributions, and effect sizes should be included in the explanation of the statistical tests (Mann-Whitney U and Fisher's exact test).

b. Reformat tables to improve readability and uniformity; some cells have uneven alignments and combine text and numerical data.

c. Figures, particularly Sankey diagrams, ought to include stand-alone captions that completely explain the contents and more lucid legends.

5. Discussion and Interpretation

a. The practical implications of the findings for laboratories at various degrees of QMS development should be further explored.

b. Strengthen the connection between the findings and the body of research on ISO 15189 implementation and QMS evolution in various resource or geographic situations.

c. Extend the limits section to include information about the findings' contextual specificity, limited dataset, and possible assessor bias.

**Do you want your identity to be public for this peer review?** For information about this choice, including consent withdrawal, please see our Privacy Policy

Reviewer #1: **Yes:** Hussein Alburkat

Reviewer #2: **Yes:** Semiu Ayinla Alayande

---

## [Author Response · Author response to Decision Letter 1]

30 Dec 2025

Dear Editor:

Thank you very much for taking the time to review our manuscript and for providing us with such valuable and constructive comments. In revising our article according to your suggestions, our team has not only gained a deeper understanding of quality management systems in medical laboratories but also developed more rigorous scientific thinking. This process has provided us with more practical directions for our future research.

If some revisions still fall short of your expectations, possibly due to our occasional misinterpretation of certain points. We sincerely aimed to reflect the growth and insights our team has gained throughout this revision process in the revised manuscript. If the revision progress has been slower than expected, we kindly ask for your understanding.

Once again, we extend our heartfelt gratitude to you and the reviewers for your careful guidance. It is with your support that the academic depth and practical value of this study have been significantly enhanced.We hope you had a wonderful holiday season and wish you all the best for the coming New Year.

Journal Requirements:

Please ensure that your manuscript meets PLOS ONE's style requirements, including those for file naming. The PLOS ONE style templates can be found at https://journals.plos.org/plosone/s/file?id=wjVg/PLOSOne_formatting_sample_main_body.pdf and  https://journals.plos.org/plosone/s/file?id=ba62/PLOSOne_formatting_sample_title_authors_affiliations.pdf

We ensure our manuscript meets PLOS ONE's style requirements.

2. We note that your Data Availability Statement is currently as follows:

“All relevant data are within the manuscript and its Supporting Information files.”

We are uploading the raw data from this study to the specified website/platform.

We have the ORCID iD .

4.Please ensure that you refer to Figures 1 and 2 in your text as, if accepted, production will need this reference to link the reader to the figure.

Thank you for the reminder. We have ensured that Figures 1 and 2 are now appropriately cited within the main text of the revised manuscript.

5.If the reviewer comments include a recommendation to cite specific previously published works, please review and evaluate these publications to determine whether they are relevant and should be cited. There is no requirement to cite these works unless the editor has indicated otherwise.

Thank you for this note.The reviewers did not recommend any specific publications for citation but good suggestions.

Reviewer #2: 1. Clarity and Language

1.There are a lot of typographical, grammatical, and formatting mistakes in the document (such as "the datas," "the potential," and "supervision/inspectionin"). Sometimes sentences are lengthy and complicated, making them harder to read. To improve clarity and flow, please have a professional scientific editor or native English speaker perform thorough English language editing.

We sincerely thank the Editors for their meticulous and valuable comments regarding the language quality of our manuscript.We have proofread and corrected typographical, grammatical, and formatting errors in the document.We have restructured all lengthy and complex sentences, breaking them into shorter sentences.We apologize for the deficiencies in the initial submission and are deeply grateful for your guidance and assistance throughout this process.

2."Double Helix" Conceptual Model

Although the "Double Helix" concept is an intriguing and inventive parallel, it is not well defined. Describe the model's theoretical underpinnings and how internal and external assessments work together as "strands" of ongoing quality improvement. Provide a schematic picture of the model to illustrate its functioning within the framework of ISO 15189 implementation.

We agree that the “Double Helix” concept, as an innovative analogy, required clearer definition and elaboration. In this revision, we have substantively addressed these points by grounding the model Fig.4 in Result section (Page9) and elaborating its function in Discussion section (Page12).1.Providing a Schematic Based on Results : Based on our analysis of the characteristics and interrelationships among NCs, root causes, and corrective actions from both internal and external assessments, we have introduced a schematic of the “Double-Helix Model” in the last Result section(Page9) . This Fig. 4 visually depicts how the internal and external assessment interlinked.2. Deepening Theoretical Explanation of the Model : we have added paragraph 7 In the Discussion section (Page12) to systematically explain the model’s core mechanism. We discuss how internal and external assessments function as complementary “strands.This collaborative interaction provides dynamic stability and a continuous upward driving force for the quality management system.

We are grateful for your guidance, which has been instrumental in enhancing the completeness and scholarly contribution of this work.

3.Methodological Details

Make clear if inclusion/exclusion criteria were used or if all nonconformities (NCs) found between 2021 and 2024 were examined. Give a more thorough explanation and rationale, preferably with references, for the formulas used to determine the Recurrence Rate and Corrective Action Effectiveness Rate. Describe the methods used to assure data reliability, such as independent reviewers and double-checking NC classification. Talk about the limits that come with researching a specific institution and how this could impact generalizability.

We have carefully considered all comments and have substantially revised the methodology section to enhance clarity, rigor, and reproducibility. The key modifications in the revised manuscript are detailed below.We have revised the "Method" section (page2,3)to provide explicit clarification. The "Method" section has been restructured into four distinct parts for enhanced clarity.Data collection(Page2): we explicitly state that the use of data was approved by the Deputy Director of the laboratory. Furthermore, it is clarified that all nonconformities (NCs) identified and recorded between 2021 and 2024 were comprehensively examined for this analysis.We have provided detailed explanations for the classification of NC types, root causes, and corrective actions. A dual-review process was employed to ensure data reliability.Data Analysis(page3): We have elaborated on the calculation methods for the two rates and provided detailed definitions for all concepts involved in the formulas.

We fully acknowledge that as a single-center study, the generalizability of our findings is subject to certain limitations. We have addressed content in the last Paragraph of the Discussion section (Page 12), which primarily includes the following aspects:1.Constraints of Institution-Specific Factors.2.Variability in Standard Implementation and Auditing, auditor competency and judgment.3.Inherent Limitations of the Methodology.we have suggested directions for future research in the manuscript, such as extending the observation period, quantifying changes in fundamental laboratory conditions and incorporating them as covariates in the analysis.To address these limitations and enhance the utility of our work, we explicitly recommend:When other laboratories apply the indicator framework and analytical methods developed in this study, they should first conduct a systematic evaluation of their own specific conditions and make necessary local adaptations. We wish that openly acknowledging these limitations and their implications does not diminish the value of our study ,but, rather, helps to clearly define its scope of application and provides a clear pathway for subsequent research. Thank you again for helping us enhance the academic rigor of this work.

4.Data Presentation and Statistics

a. When applicable, assumptions, sample distributions, and effect sizes should be included in the explanation of the statistical tests (Mann-Whitney U and Fisher's exact test).

b. Reformat tables to improve readability and uniformity; some cells have uneven alignments and combine text and numerical data.

c. Figures, particularly Sankey diagrams, ought to include stand-alone captions that completely explain the contents and more lucid legends.

a.We have carefully addressed the reviewer’s point regarding the explanation of statistical tests. Specifically, we have revised the ‘Statistical Methods’ subsection in the ‘Methods’ section (Page3)to provide a comprehensive explanation of the Mann-Whitney U test and Fisher’s exact test.The revisions explicitly include the following for each test:1.Assumptions: We state the core assumptions (e.g., independence of observations for both tests; the use of Fisher’s test when expected cell counts are <5).2.Sample Distributions: We clarify the conditions of our data that guided the test selection (e.g., use of non-parametric tests due to non-normally distributed count data and small sample sizes).3.Effect Sizes: We specify the effect size measures reported for significant results (the rank-biserial correlation coefficient for the Mann-Whitney U test and the odds ratio with its 95% confidence interval for Fisher’s exact test).These clarifications significantly enhance the methodological rigor and transparency of our study.

b.We have comprehensively revised all tables in the manuscript: text is left-aligned, and numerical data are center-aligned.We have split Table 1 (Page4) into three separate tables to present the content more clearly. Additionally, we have formatted the external assessment data in bold italics to facilitate differentiation.For the last table(Page9), we retained the classification of NCs to clearly illustrate the nature of recurring NCs. If you feel this approach is not suitable, we are very willing to further revise this table to achieve greater clarity and accuracy.

We have divided the Sankey figures into two sequential sections (Page7-8). Each set of diagrams includes separate charts for internal and external assessments.We have provided captions for both Fig. 2 and Fig. 3(Page8) .These captions clearly define the variables represented in each column, highlight the most frequent data flows and their counts, and include a brief comparative analysis between internal and external assessments.We wish these revisions significantly improve the clarity and professionalism of the figures.

5. Discussion and Interpretation

a. The practical implications of the findings for laboratories at various degrees of QMS development should be further explored.

b. Strengthen the connection between the findings and the body of research on ISO 15189 implementation and QMS evolution in various resource or geographic situations.

c. Extend the limits section to include information about the findings' contextual specificity, limited dataset, and possible assessor bias.

a.Thank you for this valuable suggestion. We have added the content to Paragraph7 of the Discussion section (Page 12).Differential Implications for Initial Stage Laboratories: For laboratories in the initial establishment phase, our results highlight the core guiding role of external assessments in identifying systemic deviations and establishing a compliance foundation, as well as the critical value of internal audits in horizontally extending supervision and consolidating improvements.

b.We have specifically cited evidence from studies in settings such as Hongkong (Paragraph2 of the Discussion section (Page 10))�Kenya and Indiain Paragraph8 of the Discussion section (Page 12)) .We wish these additions significantly deepen the practical value of our research, enabling its conclusions to provide more targeted guidance for laboratories with different levels of development and resource backgrounds.

c.To more comprehensively delineate the limitation of our study, we have made substantial expansions to Paragraph10 of the Discussion section (Page 12).

6. PLOS authors have the option to publish the peer review history of their article (what does this mean?). If published, this will include your full peer review and any attached files.

We choose “YES”

Reviewer #1:

The study is based on data from one medical laboratory, limiting generalizability. Multicentre validations needed to confirm findings across diverse settings.

To more comprehensively delineate the limitation of our study, we have made substantial expansions to Paragraph9 of the Discussion section (Page 12).

Internal assessors may have conflicts of interest or blind spots, especially in areas like documentations where they are also authors.

Thank you for raising this important and insightful point. We fully agree that internal assessors, particularly when they are also the authors of the documents under review, face potential conflicts of interest and blind spots in the evaluation of documentation and records. This is an inherent challenge of internal auditing.

We have explicitly addressed this issue in Paragraph3 of the Discussion section (Page10-11) To effectively mitigate this concern, we have proposed a comprehensive operational plan comprising four core measures:Strictly Enforce an "Author Recusal" and "Cross-Review" System

We wish this comprehensive strategic framework can effectively identify, manage, and mitigate the potential bias and blind spots of internal assessors in document review.

The training duration and rigor for internal assessors were notably less than for external assessors, possibly affecting assessment quality.

Thank you for raising this insightful point. We have incorporated an analysis of the assessors‘ potential influence into the revised discussion section.In response to this issue, we have made limitations to Paragraph9 of the Discussion section (Page 12). We explicitly state that differences may exist between internal and external assessors regarding their depth of understanding of ISO 15189 clauses and their tolerance thresholds for deviations in the quality system.We wish this addition not only directly addresses your specific concern but also enhances the methodological rigor and transparency of our study's self-reflection. We thank you for helping us strengthen this importa

---

## [Decision Letter · Decision Letter 1]

6 Jan 2026

Dear Dr. LUO,

Thank you for submitting your manuscript to PLOS ONE. After careful consideration, we feel that it has merit but does not fully meet PLOS ONE’s publication criteria as it currently stands. Therefore, we invite you to submit a revised version of the manuscript that addresses the points raised during the review process.

We look forward to receiving your revised manuscript.

Kind regards,

Arne Johannssen

Academic Editor

PLOS One

Journal Requirements:

Reviewers' comments:

Reviewer's Responses to Questions

**Comments to the Author**

Reviewer #1: All comments have been addressed

Reviewer #2: All comments have been addressed

2. Is the manuscript technically sound, and do the data support the conclusions?

Reviewer #1: Yes

Reviewer #2: Yes

3. Has the statistical analysis been performed appropriately and rigorously?

Reviewer #1: Yes

Reviewer #2: Yes

4. Have the authors made all data underlying the findings in their manuscript fully available?

Reviewer #1: Yes

Reviewer #2: Yes

5. Is the manuscript presented in an intelligible fashion and written in standard English?

Reviewer #1: Yes

Reviewer #2: Yes

Reviewer #1: The authors were able to answer all my comments and provided clear explanations to some concerns about the study. I recommend to accept the manuscript.

Reviewer #2: The manuscript now meets the standards of rigour, transparency, and contribution expected for publication, offering a meaningful and practically relevant addition to the literature on ISO 15189 implementation and quality management in medical laboratories.

**Do you want your identity to be public for this peer review?** For information about this choice, including consent withdrawal, please see our Privacy Policy

Reviewer #1: **Yes:** Dr. Hussein Alburkat

Reviewer #2: **Yes:** Semiu Ayinla Alayande

---

## [Author Response · Author response to Decision Letter 2]

7 Jan 2026

Dear Editor,

Thank you very much for taking the time to review our manuscript and providing us with timely feedback. We have carefully addressed all the comments and suggestions in the revised version.

We sincerely appreciate your guidance and support, which have enabled us to receive valuable feedback from you and the reviewers so promptly. Should any part of our revisions still fall short of your expectations, possibly due to our occasional misinterpretation of certain points, we kindly ask for your understanding, and we will continue to improve the manuscript accordingly.

Once again, we extend our heartfelt gratitude to you and the reviewers for your careful and constructive comments. The process of revising this manuscript has been a valuable opportunity for us to enhance our academic skills.

As the new year begins, we wish you a happy, healthy, and successful year.

Sincerely

Journal Requirements:

Thank you for your reminder. We have carefully reviewed all the reviewer comments and would like to clarify that they did not include specific recommendations to cite additional previously published works.

Thank you for your guidance regarding the reference list. In response to your comments, we have taken the following actions:1.We have carefully reviewed our entire reference list to ensure its completeness and accuracy.2.Each reference has been verified against the PubMed database, and none of the cited papers have been retracted.3.The formatting of all references has been updated to comply with the journal's specific style guide.

These updates have been made in the revised manuscript. Thank you again for your valuable oversight.

Reviewer comments:

Recommendation

Accept after minor editorial revision.The manuscript now meets the standards of rigour, transparency, and contribution expected for publication, offering a meaningful and practically relevant addition to the literature on ISO 15189 implementation and quality management in medical laboratories.

Thank you for your guidance.In accordance with the journal's formatting guidelines, we have made the following editorial adjustments to the manuscript:1.Paragraph Alignment: The left margin has been unified throughout the main text. For Table 4 and Table 5, due to their substantial content and necessary internal formatting to ensure clarity, the paragraph indentation within the tables differs slightly from the main body text. 2.Paragraph Formatting: We have ensured that no paragraph begins with an indentation, and a clear blank line now separates each paragraph.

We believe these formatting updates fully align with the journal's style requirements. Thank you once again for your and the reviewers' support throughout the review process.

---

## [Decision Letter · Decision Letter 2]

18 Jan 2026

The 'Double Helix' Model of Quality Monitoring:Risk Mapping of quality management system During Initial ISO 15189 Implementation in a medical laboratory

PONE-D-25-55373R2

Dear Dr. LUO,

We’re pleased to inform you that your manuscript has been judged scientifically suitable for publication and will be formally accepted for publication once it meets all outstanding technical requirements.

Kind regards,

Arne Johannssen

Academic Editor

PLOS One

Additional Editor Comments (optional):

Reviewers' comments:

Reviewer's Responses to Questions

**Comments to the Author**

Reviewer #1: All comments have been addressed

Reviewer #2: (No Response)

2. Is the manuscript technically sound, and do the data support the conclusions?

Reviewer #1: Yes

Reviewer #2: Partly

3. Has the statistical analysis been performed appropriately and rigorously?

Reviewer #1: Yes

Reviewer #2: Yes

4. Have the authors made all data underlying the findings in their manuscript fully available?

Reviewer #1: Yes

Reviewer #2: Yes

5. Is the manuscript presented in an intelligible fashion and written in standard English?

Reviewer #1: Yes

Reviewer #2: Yes

Reviewer #1: (No Response)

Reviewer #2: It is verified that the authors have completely complied with the necessary revisions after examining the reviewer/editor comments, the authors' reply letter, and the amended manuscript. The reference list has been carefully examined for accuracy and completeness, checked against PubMed to make sure no works that have been retracted are cited, and structured in compliance with the journal's style rules. Furthermore, all minor editing and formatting changes that were requested—such as paragraph alignment, spacing, and adherence to journal formatting guidelines—have been suitably applied. There are no unresolved concerns pertaining to the reviewer's remarks, and the authors' responses appropriately reflect the modifications made in the revised article.

**Do you want your identity to be public for this peer review?** For information about this choice, including consent withdrawal, please see our Privacy Policy

Reviewer #1: **Yes:** Hussein Alburkat

Reviewer #2: **Yes:** Semiu Ayinla Alayande

---

## [Editor Report · Acceptance letter]

PONE-D-25-55373R2

PLOS One

Dear Dr. LUO,

I'm pleased to inform you that your manuscript has been deemed suitable for publication in PLOS One. Congratulations! Your manuscript is now being handed over to our production team.

Kind regards,

on behalf of

Profesor Arne Johannssen

Academic Editor

PLOS One